# Prediction of Tinnitus Treatment Outcomes Based on EEG Sensors and TFI Score Using Deep Learning

**DOI:** 10.3390/s23020902

**Published:** 2023-01-12

**Authors:** Maryam Doborjeh, Xiaoxu Liu, Zohreh Doborjeh, Yuanyuan Shen, Grant Searchfield, Philip Sanders, Grace Y. Wang, Alexander Sumich, Wei Qi Yan

**Affiliations:** 1Knowledge Engineering and Discovery Research Institute (KEDRI), School of Engineering Computer and Mathematical Sciences, Auckland University of Technology, Auckland 1010, New Zealand; 2Centre for Robotics & Vision (CeRV), Auckland University of Technology, Auckland 1010, New Zealand; 3Eisdell Moore Centre, Audiology, School of Population Health, The University of Auckland, Auckland 1010, New Zealand; 4School of Psychology, The University of Waikato, Hamilton 3216, New Zealand; 5School of Psychology and Wellbeing, University of Southern Queensland, Darling Heights, QLD 4350, Australia; 6Centre for Health Research, University of Southern Queensland, Darling Heights, QLD 4350, Australia; 7NTU Psychology, Nottingham Trent University, Nottingham NG1 4FQ, UK

**Keywords:** tinnitus, artificial intelligence, EEG, prediction, TFI, functional connectivity, deep learning, digital health

## Abstract

Tinnitus is a hearing disorder that is characterized by the perception of sounds in the absence of an external source. Currently, there is no pharmaceutical cure for tinnitus, however, multiple therapies and interventions have been developed that improve or control associated distress and anxiety. We propose a new Artificial Intelligence (AI) algorithm as a digital prognostic health system that models electroencephalographic (EEG) data in order to predict patients’ responses to tinnitus therapies. The EEG data was collected from patients prior to treatment and 3-months following a sound-based therapy. Feature selection techniques were utilised to identify predictive EEG variables with the best accuracy. The patients’ EEG features from both the frequency and functional connectivity domains were entered as inputs that carry knowledge extracted from EEG into AI algorithms for training and predicting therapy outcomes. The AI models differentiated the patients’ outcomes into either therapy responder or non-responder, as defined by their Tinnitus Functional Index (TFI) scores, with accuracies ranging from 98%–100%. Our findings demonstrate the potential use of AI, including deep learning, for predicting therapy outcomes in tinnitus. The research suggests an optimal configuration of the EEG sensors that are involved in measuring brain functional changes in response to tinnitus treatments. It identified which EEG electrodes are the most informative sensors and how the EEG frequency and functional connectivity can better classify patients into the responder and non-responder groups. This has potential for real-time monitoring of patient therapy outcomes at home.

## 1. Introduction

Tinnitus is characterized by the perception of sound in the absence of an external sound source. Although the effects of tinnitus vary from person to person, most patients suffer from varying degrees of fatigue, stress, sleep problems, difficult in concentrating, memory loss, anxiety, and irritability [1]. The most common cause of tinnitus is exposure to loud noise, while other causes include ototoxic drugs (e.g., aspirin), head injury, ear infection, or other diseases, such as diabetes [2].

The acute effect is injury to the cochlea and auditory nerve, following auditory plasticity, tinnitus-related effects centralise over time. Due to the heterogeneity of tinnitus, treatment response varies. Clinical judgements about the likelihood of treatment response are currently difficult, and patients are often switched through several treatments before finding an effective intervention. AI may help in personalising the prescription approach to tinnitus by indicating those interventions most likely to be effective for any given patient. This would improve clinical efficiency, reducing patient distress and optimising benefits. Although several AI methods have been applied to brain data in order to classify (diagnosis) tinnitus (e.g., support vector machine, SVM; multilayer perceptron, MLP; logistic regression; and naive Bayes learning) [3,4,5,6], there is an absence of research on the early prediction of symptom outcomes (prognosis). Prediction of response might be possible by applying machine learning methods to diverse spatio-temporal brain data (e.g., electroencephalogram, EEG [7]; magnetic resonance imaging, MRI [8]; functional MRI [9]; functional near-infrared spectroscopy, fNIRS [10]; and magnetoencephalography, MEG [11]), clinical and behavioural measures (e.g., hearing thresholds, sleep quality), neuropsychological tests (e.g., memory); and/or cardiovascular measurements (e.g., heart rate variability) [12].

The current study focuses is on EEG, which allows for high temporal-resolution recording of cortical electrical activity via electrodes positioned over the scalp [13,14]. By using machine learning to model high-dimensional EEG data, feature selection methods can be applied to remove irrelevant data, hence, a smaller number of data features which can reduce the model complexity. This helps ro prevent overfitting and improves learning performance by promoting generalization [15,16,17,18]. Sub-sampling in the time domain or frequency domain of EEG signals is a common method for feature extraction which is broadly utilized in neurological diagnosis. Combining EEG features from the temporal and frequency domains leads to increased accuracy in pattern classification [13]. Reference [19] proposed converting EEG features from the temporal domain to the frequency domain using Fast Fourier Transform (FFT) and illustrated them as a series of multispectral images of brain topology. These images were then used to train machine learning algorithms, including deep learning neural networks, to learn from the robust representation of image sequences. One of the most popular methods in machine learning is based on deep neural networks. Deep neural networks (DNN) are computational models that simulate the way neurons process information. They can identify patterns in data and make predictions based on those patterns. A variety of architectures of DNN have been proposed so far, including the Convolutional Neural Network (CNN) [20]. The CNN is a feed-forward neural network with artificial neurons that respond to surrounding units, and it is excellent for image processing. The CNN consists of one fully-connected layer (first layer), one or more convolutional layers, as well as associative weights and a pooling layers. CNNs can also be trained using back-propagation algorithms. Compared to other deep neural networks, convolutional neural networks require fewer parameters to be considered, making them an attractive structure for deep learning [21]. Modelling of high-dimensional EEG data using machine learning methods has been conducted for the diagnosis of various neurological diseases, including epilepsy and seizures [22]. Reference [23] proposed a fuzzy decision tree (FDT) classifier for epileptic seizure detection which achieved 99.5% accuracy. Despite the existence of several successful EEG applications in the health field, there are still limited studies on modelling EEG for the prediction of responses to tinnitus treatment. Most of the EEG studies are based on the classification of raw EEG signals using machine learning. Limited investigation has been conducted on using dynamic functional connectivity and dynamic frequency images as deep learning inputs for the prediction of tinnitus treatment outcomes.

In this paper we explain the methodology of our predictive models as well as the feature selection methods used for improving the performance of the models. We report the predicted results of treatment outcome classification (i.e., responder, non-responder). A comparative analysis using different feature selection methods is reported on EEG data in the frequency and functional connectivity domains.

## 2. Methodology

This research aims to develop an AI-based system for predicting the change in patients’ severity of tinnitus over a period of treatment using computational models of brain data collected prior to and following a tinnitus treatment (the EEG data is explained in [24]). The primary measure of tinnitus severity used was the Tinnitus Functional Index (TFI), a questionnaire designed to measure the impact of tinnitus on various aspects of life (e.g., sleep, communication, and quality of life) [25]. The methods are illustrated in Figure 1 and include modelling EEG data in the frequency domain and functional connectivity networks applied as inputs to artificial intelligence algorithms (neural networks) for extracting patterns and performing prediction of the tinnitus treatment outcomes.

The clinical impact of the proposed methodology is applying AI for the early prediction of response to tinnitus treatment when only baseline data is used. This would allow for an optimal selection of treatment options for patients through the prediction of response to each treatment. We also identified a subset of EEG sensors as informative features that increase the prediction accuracy and make the further development of wearable diagnosis and prognosis AI tools more effective. Sensors have seldom been used clinically in relation to sensory disorders. The application of sensor technology to tinnitus is novel, and the algorithm and its application are innovative approaches to this complex sensory disorder. The literature has clearly identified the need for biomarkers, and this is amongst the first studies looking towards effective wearable solutions in our field.

### 2.1. Datasets

We used two datasets (EEG, behavioural) from patients with tinnitus (n = 8 shown in Figure 1a) at baseline (pre-treatment) and after the treatment (a follow up after 3 months). The behavioural data recorded 19 features, including cognitive, psychological, and the TFI scores collected at each follow-up, as listed in Table 1. The TFI score in behavioural data was used to categorise patients into two groups (responders and non-responders), according to the level of change in their the TFI scores between the pre- and post-treatment phases. The responsiveness of the TFI to treatment-related change was evaluated by examining the change in baseline test–retest scores calculated as TFI change = post TFI − baseline TFI. The smallest detectable change was determined from the variance in these measures and was found to be 4.8. Therefore, the responder group encompasses individuals with TFI changes ≥ 4.8, while the non-responder group encapsulates individuals with TFI changes < 4.8. This labelling criterion was also validated in the literature for New Zealand data [26]. Among 8 patients, 4 of them were labelled as responders, while the other 4 patients were labelled as non-responder according to their TFI score changes.

EEG was recorded in an electrically shielded and sound-treated booth (ISO 8253–1:2010) from 64 BioSemi active Ag/AgCl recording electrodes. Electrode locations corresponded to the extended international 10/20 system. Electrodes were attached to a fitted BioSemi head cap. Parker Signa gel was applied at each electrode site to ensure reliable conductivity between electrode and scalp. Continuous EEG signals were recorded on a Dell Optiplex 7040 desktop computer at an 8192 Hz sample rate with a 64-channel BioSemi ActiveTwo system referenced to the common mode sense active electrode and grounded to the driven right leg passive electrode. The EEG signals were down-sampled to 256 Hz. The temporal lengths of these EEG signals were segmented into multiple intervals of 256 time-points (i.e., 1 s duration). This resulted in 6642 EEG samples, which were used for training deep neural networks to predict the patient’s tinnitus treatment outcomes (class labels: responder vs. non-responder defined with respect to the changes in their TFI score after 3 months of treatment). The class labels of the EEG samples correspond to the TFI changes in the behavioural data of the same patients.

Tinnitus severity numerical scales (TSNS) have been widely used to assess tinnitus severity and have demonstrated good test–retest reliability and concordance with other participative measures of tinnitus. Participants were asked how much a problem their tinnitus was (0 not a problem−5 very big problem). Numeric rating scales were used to measure tinnitus perception along five dimensions: how strong, intrusive, uncomfortable, unpleasant the tinnitus signal was, and how easy it was to ignore the tinnitus signal (0–10 rating, 0 not a problem−10 extreme problem) [27]. The literature suggested statistical results that proved the validity of this scale [28,29].

### 2.2. Psychological Function

Depression, Anxiety, and Scale (DASS) [30] is a validated, widely used self-reporting instrument that measures the dimensions of depression, anxiety, and stress for use in both clinical and nonclinical populations. Participants rate 21 items (7 for each subscale) on a 4-point scale of how much each statement applies to them. The Positive and Negative Affect Schedule (PANAS) [31] is a self-reporting questionnaire that consists of a list of 10 positive and 10 negative affective adjectives, each rated on a 5-point scale (not at all–very much).

### 2.3. Prediction of Treatment Outcomes Using EEG Data and Neural Networks

In the current research, we applied neural networks for the classification of EEG data (only baseline EEG) in order to predict patients’ treatment outcomes (responder and non-responder groups) labelled after 3 months of tinnitus sound treatment. To this end, a tinnitus avatar was first generated that was identical to the individual with tinnitus. This sound was then morphed slowly over time until it was identical to an environmental sound. Within the field of virtual reality, one study has attempted to synthesize an auditory replica of tinnitus. However, this relied purely on auditory thresholds, tinnitus pitch, and participant preference judgments [32].

In the current study, the computational AI model has the potential to distinguish the baseline EEG patterns of patients who are likely to respond to the tinnitus treatment over time. In order to identify different types of predictive patterns from EEG data, we considered two forms of EEG-driven information as inputs to the neural networks: (1) frequency domain (presented in Section 2.3.1) and (2) EEG functional connectivity (presented in Section 2.3.2).

#### 2.3.1. EEG Class Prediction on Frequency Domain

In this section, we demonstrate EEG data modelling via deep neural networks for the prediction of treatment outcomes. Here, the EEG modelling is based on the frequency domain. A FFT (fast Fourier transform) was applied to transform each of the EEG samples into the frequency domain image. As described in Equation (1) and Reference [33], we first created EEG samples with a 1 s duration. As explained in Section 2.1, 6642 frequency images were generated. To make use of meaningful data, we selected features on the most prominent frequency bands (beta 14–30 HZ, alpha 8–14 HZ, and theta 4–8 HZ) as the analysis objects. The mean (across each 1 s epoch) of the absolute values for each of these frequencies was calculated at each electrode site.
(1)$$X_{k}=\sum_{n=0}^{N-1}x_{n}e^{-i2\pi k\frac{n}{N}}\;k=\;0,\;1,\;2,\ldots .,\;N\;-\;1$$
where $X_{k}$ is the FFT coefficients, N is the total number of input EEG samples, and n is total number of points in FFT. The frequency samples were then spatially mapped onto 64 electrodes, based on the international 10/20 system co-ordinates, and shown in 2D surface frequency images. Specifically, a grid data function based on Delaunay triangulation was used to interpolate the data points and generate frequency domain images. After the triangulation was completed, any points in the triangle area were interpolated according to the value of each triangle vertex. This process was repeated for all frequency bands of interest, resulting in three topographic maps corresponding to each frequency band (Figure 2). The amplitudes of the three frequency bands were then averaged and passed to a single topographically based frequency image which dynamically changes over time when streaming temporal EEG data [34,35].

Data from the sequence of 1 s epochs was used as a temporal input into the neural networks for training. In this experiment, because the EEG datasets were presented in the form of images, we employed a CNN in deep learning [36], commonly used in image classification. Such CNNs are inspired by the visual cortex, where the firing rate of every sensory neuron is affected by a specific region in the retina, called the neuron’s receptive field. CNNs consist of three main layers: the input layer, the feature learning layer, and the classifier layer. Each of these has several sub-layers. Through a convolution procedure, every region of neurons (receptive field) from layer i was connected to one neuron in layer i+1, which resulted in extracting abstractions (informative features from data) from layer i and transferring them to the next layer. CNNs use activation functions, therefore, they can solve non-linear classification tasks and have enabled advancements in computer vision systems, including image classification [22], image segmentation [37,38], and object detection [39]. In the current study, CNNs were applied for the classification of EEG frequency images into responder and non-responder groups. As shown in Figure 3a, 2D images with a size of 32 × 32 pixels were harnessed as EEG frequency inputs into the CNN model. We employed ReLU as the activation function in the convolutional layers. The softmax was applied in the last activation layer for final classification and prediction.

#### 2.3.2. EEG Class Prediction Based on Functional Connectivity

In this section, EEG data was transformed into functional connectivity representing the relationship between different cortical regions over time. Graph theory was applied to each EEG frequency domain in order to convert it into a graph-based representation of the frequency domain, shown in Figure 3b. We employed squared coherence ($Coh_{xy})$ to compute the degree of correlation between every two EEG channels x and y in the frequency domain, as shown in Equation (2). Here, the three frequency bands of theta, alpha and beta were used.
(2)$$Coh_{xy}=\frac{{\left|P_{xy}\left(f\right)\right|}^{2}}{P_{xx}\left(f\right)\cdot P_{yy}\left(f\right)}$$
where *P_xy_* represents the cross-spectral density of the signal *x* and *y*. Parameters *P_xx_* and *P_yy_* show the power spectral density. The generated EEG functional connectivity graphs were used as inputs to train a deep neural network. Hereafter, we employed MLP as the classifier.

Figure 4 shows an example of EEG functional connectivity graphs generated from one EEG sample (one second recording) for eight patients at two stages (before and after treatment). The results demonstrate that both the responder and non-responder groups have increased functional connectivity after receiving treatment, with the responder showing a greater increment in their EFC in the cortical areas measured by F1, Fz, F2, FC1, FCz, FC2, C1, Cz, and C2 electrodes. Functional connectivity in non-responder patients increased more significantly after treatment and was spread across all electrodes.

## 3. Results of Tinnitus Outcome Prediction Using EEG Data

### 3.1. Prediction Treatment Outcomes Based on EEG Frequency Domain

As explained in Section 2.1, we created 6642 EEG samples, each with a recording length of 1 s with 256 data points. These samples were transformed into frequency images (explained in Section 2.3.1) and employed as inputs to the CNN model. Before training the CNN model, we visualised an example of frequency images of 5 s of EEG data related to two patients from the non-responder group (in Figure 5) and the responder group (in Figure 6).

For the prediction of tinnitus treatment outcomes, only the baseline EEG data from patients were used to train the CNN model. The class label information for the baseline EEG data was defined according to the patients’ TFI scores after treatment. The trained CNN model was then tested using patients’ baseline EEG data, which were excluded from the training to predict whether the patients are likely to be classified as non-responders or responders. For the training and testing of the CNN, the EEG samples were split into training and test datasets in a ratio of 8:2, meaning that there were 5314 training samples and 1328 test samples. As reported in Table 2, the prediction accuracy of the non-responder group is 99.07%, while the responder group was predicted with 98.86% accuracy. Figure 7 shows the accuracy and the loss curves during CNN training. In the experiments, we used different epoch numbers (from 20–100) and learning rates (0.01, 0.001, 0.0001, and 0.00001) with the aim of finding the optimal parameters that generate the highest accuracy. After 80 epochs of training with learning rate of 0.00001, the overall accuracy of the model was reached to 98.94%.

To assure that the CNN model was robust and stable, we also applied *k*-fold cross validation for training and testing the CNN model. The results of different numbers of fold validation are reported in Table 3, which illustrates the model robustness.

### 3.2. Prediction of Treatment Outcomes Based on EEG Functional Connectivity

In our experiment, the raw EEG signals were transformed into functional connectivity graphs with respect to the coherence measured between every pair of EEG channels as defined in Equation (1). The functional connectivity is a dynamic graph in which the nodes are the EEG channels, and the arcs demonstrate the squared coherence between the channels. The graph was updated with respect to the changes in EEG signal over time, thereby, generating time series information that can be used for the training of deep learning neural networks based on MLP.

Figure 8 visualises the functional connectivity graphs of 5 s EEG data related to two patients from the responder and non-responder classes. Here, we set the coherence visualisation threshold for the graph arcs to 0.75. This means that if the squared coherence between two EEG channels was greater than 0.75, then there was a connection between the two nodes in the functional connectivity graph.

Figure 8 shows functional connectivity graphs generated from 80 s of EEG data from non-responsive and responsive patients before and after treatment. This can be used to better understand the functional changes as a result of treatment. For instance, the functional connectivity increased to a wider area of the brain, including Fpz, AFz, Fz, FCz, AF3, and AF4) after treatment in the responder group. This increment, however, was not seen in the non-responder group.

We applied MLP as the model to predict classes (treatment outcomes) based on the EEG functional connectivity graphs, each with a size of 64*64 cells (representing pairwise correlations between 64 EEG channels). We applied 8:2 training and testing split for the classification reported in Table 4. The total accuracy was 99.41% (the non-responder group accuracy was 99.28%, and the responder group accuracy was 99.50%).

## 4. Feature Selection for Identification of Tinnitus Predictive EEG Variables

The experiments for the prediction of tinnitus treatment were conducted using all 64 EEG channels. In order to reduce the dimensionality of the data and the computational model complexity, those EEG channels that were not significant for tinnitus diagnosis were detected and eliminated. We applied a variety feature selection method to determine the most important EEG variables affected by tinnitus treatment. Here, we developed a new approach, called greatest change channel selection (GCCS), for measuring the importance of the EEG channels in classifying the EEG samples to pre- and post-treatment states. This feature-selection method focused on finding the channels with the greatest changes caused by treatment. Equation (3) calculates $E_{c_{i}}$, which is the average degree of treatment effect on each EEG channel denoted by $C_{i}$.
(3)$$E_{c_{i}}=\frac{\sum_{j=0}^{M}\left|A_{C_{ij}}-A_{C_{ij}}^{\prime }\right|}{M}$$
where *M* is the number of patients, *i* is the *i-th* channel, and *j* is the *j-th* patient. $A_{C_{ij}}$ is the pre-treatment amplitude of the patient *j* in EEG channel *i*. $A_{C_{ij}}^{\prime }$ is the post-treatment amplitude of the same patient *j* in the same EEG channel *i*. Here the range of I is [1–64] and j is [1–8].

For the GCCS feature selection method, we calculated the changes in the amplitude of each EEG channel before and after treatment. After calculating the average degree of treatment effect $\left(E_{c_{i}}\right)$ for all 64 channels in all patients, we identified the top 30 EEG channels that demonstrated the greatest average of $E_{c_{i}}$ across all patients. We also tested five other feature selection methods, including F_Regression (FR), random forest (RF), ExtraTrees (ET), and RFE. The feature selection methods are called model-based ranking. We first calculated the average amplitude on each EEG channel (across patients) before and after treatment and obtained two (before and after treatment) 64 × 8 matrices, in which 64 represents the number of EEG channels, and 8 represents the number of patients. Then, another average was taken across 8 patients on each channel, thus generating two 64 × 1 matrices, one represents the EEG status before the treatment (called before-matrix) and one represents the EEG status after the treatment (called after-matrix). We entered these two 64 × 1 matrices into the feature-selection methods as hereafter described.

For the FR method, the model returned the F-statistic and *p*-values as the criteria for ranking the importance of the channels. The larger the *p*-values, the higher the importance of the corresponding channel. This was calculated by measuring the correlation between the before-matrix data and the after-matrix data based on r-regression. Then, the cross correlation was converted to an F-score and then to a *p*-value.

The RF method ranked the importance of channels for each tree according to the impurity, which was calculated based on variance.

The ET method ranked the importance of the channels according to the impurity-based feature importance. The higher the rank, the more important the channel. The importance of a channel was computed as the (normalized) total changes in the value of the channels.

The RFE method determined the importance of each channel through the feature importance attribute returned by the model. Then, the least-important channels were removed from the current set of channels. This step was repeated recursively on the channel set until the required number of channels was finally reached.

Table 5 shows the top 30 EEG channels selected by the aforementioned feature selection methods as well as our proposed GCCS method. Table 6 demonstrates that the GCCS method resulted in the highest prediction accuracy (99.47%), followed by F_Regression (99.39%) and RFE (99.09%).

## 5. Conclusions

AI diagnostic prediction is an extremely promising new field for the treatment of tinnitus. Due to the richness and variety of treatment methods for tinnitus and the variation in tinnitus treatment outcomes amongst individuals, the use of AI diagnosis and prediction may lead to a better treatment plan. This shortens the treatment time and allows patients to receive more targeted and personalized treatment. The goal of this study was to build artificial intelligence models to predict the outcome of tinnitus treatment. This research applied different neural networks (including CNN and MLP) to learn from patients’ EEG data for predicting their treatment outcome. The neural networks modelled EEG frequency features and functional connectivity features, and they resulted in up to 99% accuracy of for prediction of patients who were responders or non-responders to treatment. To the best of our knowledge, no other method for predicting the treatment outcome of tinnitus by analysing frequency features and functional connectivity has been published. Moreover, there is a lack of EEG studies investigating tinnitus treatment. Compared with previous works using time-domain features as the object of analysis [40,41,42,43], we use frequency-domain features and functional-connectivity features to identify signal changes or patterns with higher feasibility, rather than just observing a single time domain information. As time-domain features depend on the nature of the signal, using frequency-domain features and functional-connectivity features provides stronger discriminative power. Compared with works that only use the frequency-domain information of a single frequency band for analysis [43], our work merges three frequency bands in order to obtain a low-dimensional representation, which enriches the features and reduces the computational cost of the model. Furthermore, we extracted correlations (functional connectivity features) between the signals of different sensors and found that these functional connectivity features can be linked to treatment outcomes in tinnitus patients.

So far, the highest accuracy of our predictions comes from the frequency-domain model, at 99.52%. At the same time, the functional connectivity-based model also performed very well, with an accuracy of 99.41%. In the process of model training, we used 70% of the data for training and the remaining 30% for network performance testing. We also conducted 5-fold cross-validation on the model, and the results were all above 90% accuracy. This suggest that our model is sufficiently robust. This is strong evidence that EEG signal analysis by AI models can reliably predict the outcome of tinnitus treatment. In addition, we applied five conventional feature-selection techniques (FR, RF, ET, and RFE) to identify the top predictive EEG variables that lead to increasing the prediction accuracy. We also proposed a new feature-selection approach, called GCCS, which resulted in the best prediction accuracy when compared to other methods. The GCCS method identified FC3, P8, P4, T8, and CP5 as the main predictive EEG variables.

This research also enabled visualisation of the topography of the human brain for frequency-domain features of the EEG, and visualisation of the functional connectivity of the human brain. The frequency-domain visualization (Figure 5 and Figure 6) allowed for investigation of the EEG amplitude changes in theta, alpha, and beta bands as a result of tinnitus treatment. According to the functional connectivity visualization, both the responder and non-responder groups have increased functional connectivity after receiving treatment, with the responder showing a greater increment in their connectivity in the cortical areas measured by F1, Fz, F2, FC1, FCz, FC2, C1, Cz, and C2 electrodes. Functional connectivity in non-responsive patients increased more significantly after treatment and was spread across all electrodes.

The research proved that within EEG data, both frequency and functional connectivity contain significant information showing brain changes as a response to treatment. Each of the frequency images and the functional connectivity graphs was generated using a 1 s EEG signal (256 time points), and they were used as inputs to deep learning and achieved a greater accuracy of prediction compared to using the whole time points of raw EEG signals.

For the future, a real-time prognostic digital health system is planned to be developed based on a small number of EEG variables (selected through feature selection) for the potential design of a wearable system for patients as home. To this end, a more robust AI model needs to be trained using more EEG data from tinnitus patients. This will allow us to identify a generalised group of EEG channels associated with tinnitus and the effect of treatment.

## Figures and Tables

**Figure 1 sensors-23-00902-f001:**
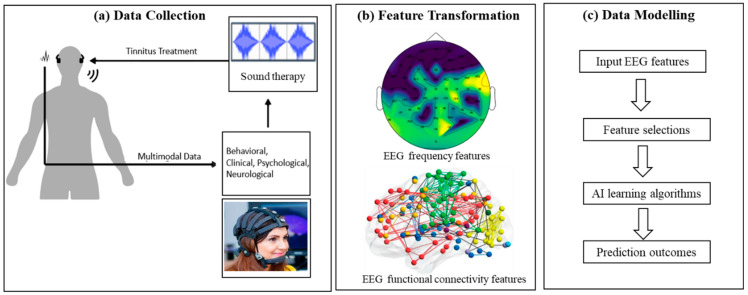
The protocol of the study: (**a**) seventeen chronic tinnitus patients undertook tinnitus treatment, and their multimodal data were collected. EEG data were collected from each patient at baseline (pre-treatment) and at three follow-up times. (**b**) The EEG features were transformed into frequency and functional connectivity domains and used as input data to the neural network algorithms. (**c**) Procedure for data modelling and predicting the treatment outcomes (responder and non-responder).

**Figure 2 sensors-23-00902-f002:**
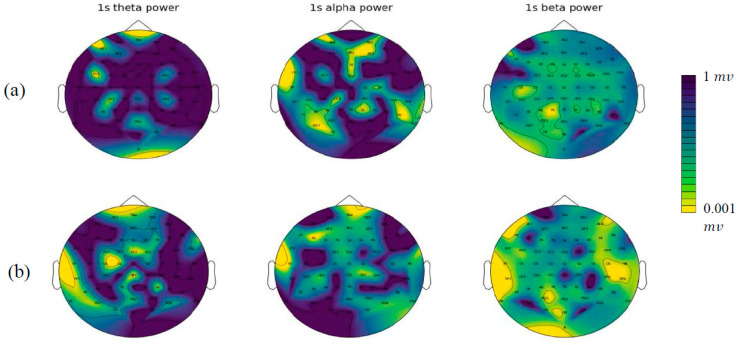
An example of one time-frame image of three frequency bands from one patient at baseline (**a**) and after treatment (**b**). These colour maps are the power topographies of each frequency band. The colours from blue to yellow represent powers from weak to strong.

**Figure 3 sensors-23-00902-f003:**
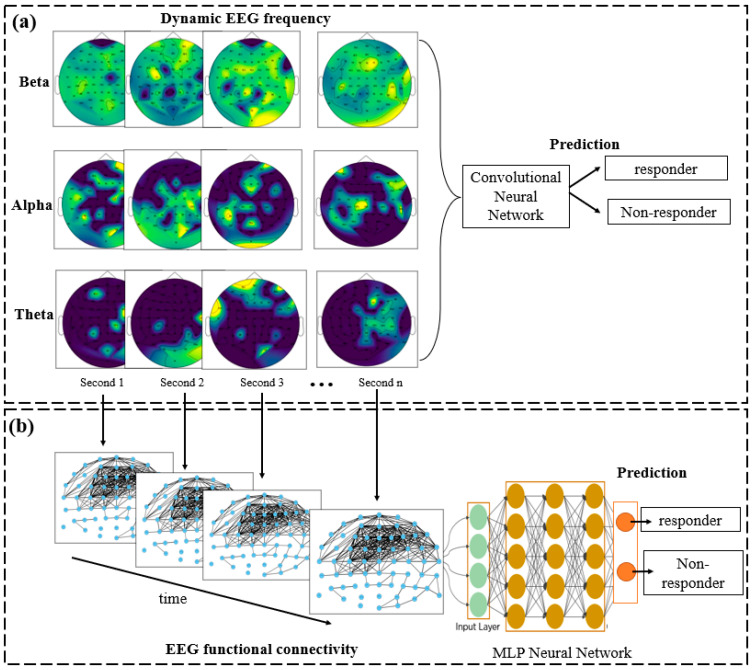
The proposed deep learning-based architecture for modelling of EEG data and the prediction of treatment outcomes using (a) dynamic frequency images via CNN, and (**b**) functional connectivity via MLP.

**Figure 4 sensors-23-00902-f004:**
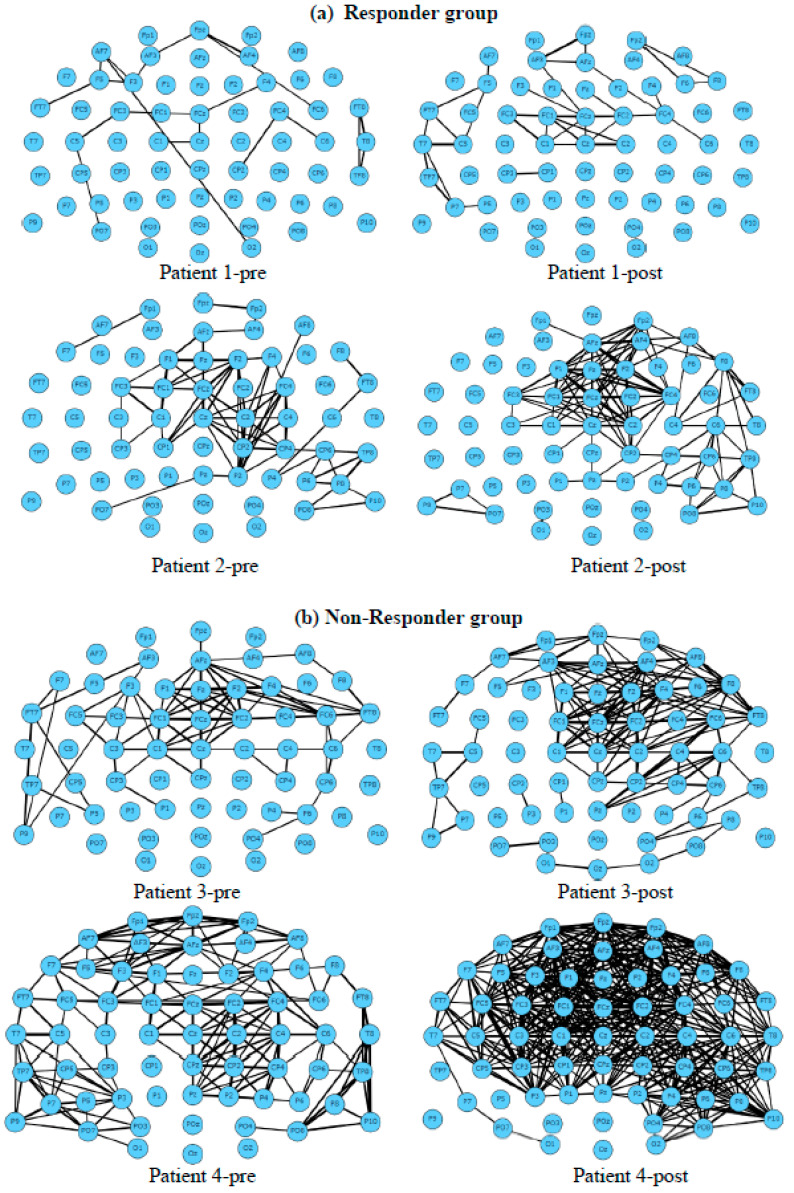
EEG functional connectivity maps were generated for four patients using EEG data from baseline (pre-treatment) and post-treatment. (**a**) Maps from 2 patients from the responder group. (**b**) Maps from 2 patients from the non- responder group. Nodes represent the EEG channels, while the lines represent the correlation between every two channels calculated using Equation (1). Figure 5 and Figure 6 demonstrate the frequency images (theta, alpha, and beta) from two randomly selected patients belonging to non-responder and responder groups, respectively, that were generated from 80 s of EEG. These frequency images provide an opportunity to further investigate the brain areas affected by the tinnitus therapy. For example, Figure 5 shows that for a patient from the non-responder group, the power of alpha frequency was increasing over time across most of the brain regions before treatment, and a similar pattern was seen after treatment. On the other hand, Figure 6 demonstrates that theta and alpha were shown to be more prominent after treatment in the responder patient. These patterns of changes were used as inputs for the training of deep neural networks to distinguish who is likely to respond to treatment.

**Figure 5 sensors-23-00902-f005:**
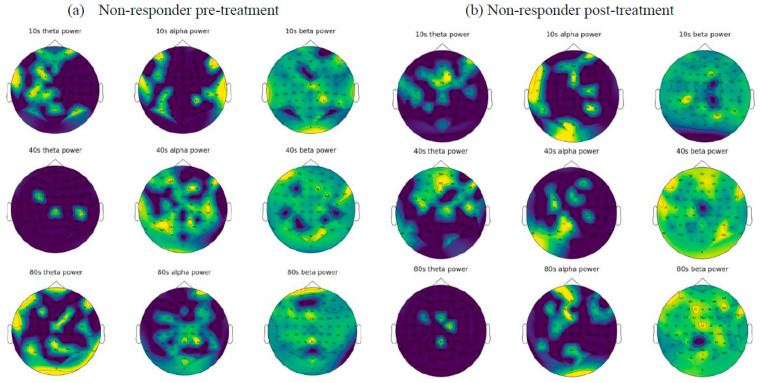
Visualization of the EEG frequency domain features from a randomly selected non-responsive patient before (**a**) and after (**b**) treatment.

**Figure 6 sensors-23-00902-f006:**
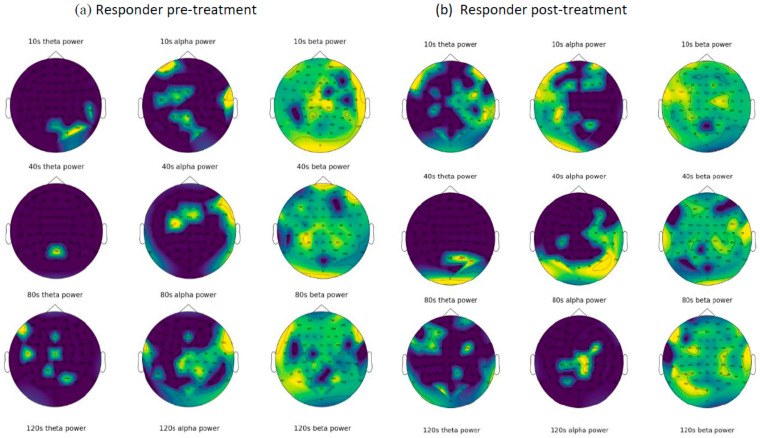
The visualization EEG frequency domain features from a randomly selected responsive patient before (**a**) and after (**b**) treatment.

**Figure 7 sensors-23-00902-f007:**
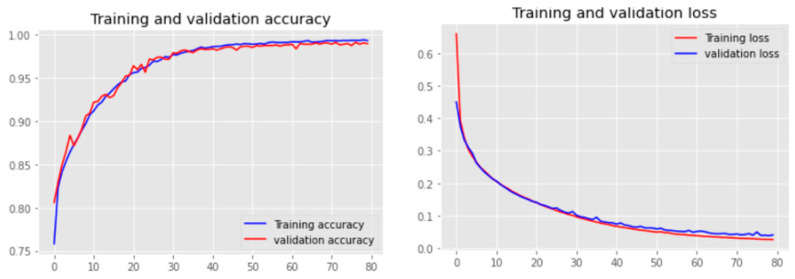
Accuracy and loss curves during the training.

**Figure 8 sensors-23-00902-f008:**
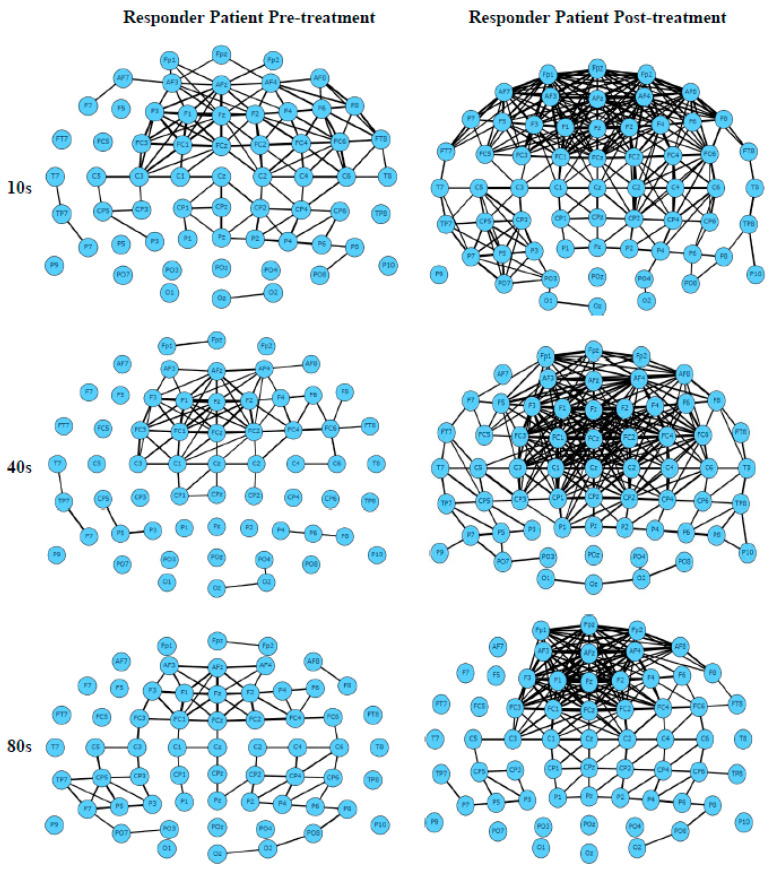
The functional connectivity graphs generated from 80 s of EEG data (averaged across patients) in the non-responder and responder groups, pre- and post-treatment.

**Table 1 sensors-23-00902-t001:** The data features including cognitive, psychological, and TFI scores.

Test	General	TSNS	DASS	PANAS
Features	- Intrusiveness - Sense of Control - Concentration - Sleep quality - Auditory - Relaxation - Quality of Life - Emotion	- Overall (1 = Not a problem, 2 = Small, 3 = Moderate, 4 = Big, 5 = Very big) - Strong - Uncomfortable - Annoying - Ignore Unpleasant	- Depression - Anxiety - Stress	- Positive Affect - Negative Affect
Total score	8	6	3	2

**Table 2 sensors-23-00902-t002:** Confusion table for the prediction of the non-responder class and the responder class using the CNN model which is trained and tested by pre-treatment EEG frequency images to predict the outcome after the treatment (training and testing sample ratio is 8:2).

Prediction Label\True Label	Non-Responder	Responder	Accuracy	Specificity	Sensitivity
Non-responder	533	9	99%	98.9%	98.34%
Responder	5	782			

**Table 3 sensors-23-00902-t003:** The accuracy of the prediction of non-responder and responder groups using CNN via k-fold cross validations.

Folds	5-Fold	6-Fold	7-Fold	8-Fold
Accuracy	99.37%	99.52%	98.13%	98.79%

**Table 4 sensors-23-00902-t004:** The predictive accuracies using MLP classifies based on EEG functional connectivity features.

Prediction Label\True Label	Non-Responder	Responder	Accuracy	Specificity	Sensitivity
Non-responder	138	1	99.28%	99.5%	99.5%
Responder	1	202	99.50%		
Total	139	203	99.41%		

**Table 5 sensors-23-00902-t005:** Top 30 most-important EEG channels as selected by different feature selection methods, including F_Regression (FR), random forest (RF), ExtraTrees (ET), RFE, and our proposed GCCS method.

Rank\Method	FR	ET	RFE	RF	GCCS
1	F6	F6	Fp1	Fp1	FC3
2	P2	P2	AF7	F6	P8
3	FC5	FT8	F3	TP7	P4
4	CP3	T8	F5	AF8	T8
5	Fp2	Fp2	F7	P2	CP5
6	Fp1	FC5	FT7	CP3	F3
7	AF8	Fp1	FC5	CPz	CPz
8	P8	T7	C5	F3	F6
9	TP7	CPz	T7	FC1	Fz
10	Oz	F2	TP7	Fpz	FT7
11	CP6	CP3	CP5	FC2	Fpz
12	O2	F1	CP3	P5	O2
13	O1	O2	CP1	F4	Poz
14	CPz	Pz	P7	P8	TP7
15	CP1	Fpz	P9	Oz	FT8
16	F4	C2	Iz	FT8	FC4
17	FC3	P6	Oz	PO4	F4
18	P5	P5	Pz	F1	AF8
19	F3	C5	CPz	FC5	Oz
20	PO8	CP6	Fpz	AF3	FC5
21	C2	F3	AF8	Fp2	AF4
22	TP8	Fz	F2	CP4	C4
23	P7	AFz	F6	AF4	F1
24	Fpz	AF4	F8	T7	AF7
25	C5	FC3	C6	Fz	AF3
26	PO7	Iz	CP6	P1	T7
27	FC6	AF8	P2	C2	C1
28	PO3	PO4	P8	Cz	CP6
29	C6	CP5	P10	F5	P5
30	F8	FT7	O2	C3	PO8

**Table 6 sensors-23-00902-t006:** The prediction accuracy of CNN using smaller sets of EEG channels (top 10, top 20, and top 30) selected via our proposed GCCS method and compared with five other methods including RFE, F_Regression (FR), random forest (RF), ExtraTrees (ET), and RFE.

Feature Selection Technique\the Number of Channels	Top10	Top20	Top30	Model Configuration
FR	90.86	97.96	99.39	Center = True Force_finite = True
RF	89.46	97.29	97.74	max_features: Auto
ET	94.80	98.11	98.04	n_estimators = 100
RFE	93.52	96.99	99.09	criterion: squared_error
GCCS	95.71	97.59	99.47	Not Applicable

## Data Availability

Data collection was approved by the University of Auckland Human Participants Ethics Committee.

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
