# Peer review of "Prediction of Tinnitus Treatment Outcomes Based on EEG Sensors and TFI Score Using Deep Learning"

_sensors, 2023, doi:10.3390/s23020902_

Round 1

Reviewer 1 Report

THe main weaknesses of the manuscript are:

1) what are the advancement respect to the state of art (including previous manuscript published by authors). It would be easier for a reviewer to have the overall results produced by authors cited in chronological order.

2) What is the real advancements in the field of the Journal. SInce there are no novelties in the filed of sensors one would expect some advancements in the field of signal processing.

3) anothe rpoint that should be in depth discussed is the potential clinical impact of these results. 

Author Response

Response to reviewers’ comments and summary of changes   

First of all, we would like to thank the reviewers for their very professional and time-consuming work to make this paper better organised and clearer, so that it will have a much stronger impact on the future research in our community.  It is novel to apply sensor technology to tinnitus and the algorithm and its application are innovative approaches to this complex sensory disorder.  We believe that our paper is a pioneering study in modelling EEG frequency domain and functional connectivity to predict response to tinnitus treatments.  

Below are the questions raised by the reviewers and our revision made in response.   

Reviewer #1:

The main weaknesses of the manuscript are:

  • what are the advancement respect to the state of art (including previous manuscript published by authors). It would be easier for a reviewer to have the overall results produced by authors cited in chronological order.

To the best of our knowledge, no method to predict tinnitus treatment outcomes through analysing frequency features and functional connectivity has been published. Moreover, there is still limited studies on modelling EEG for investigating response to tinnitus treatment. This research is based on using deep neural networks trained on extracted patterns from EEG data that are 1) dynamic frequency images and 2) dynamic functional connectivity, applied for the first time to predict tinnitus treatment outcomes. Compared the previous research in which we designed a brain-inspired spiking neural network model to classify raw EEG signals, here the extracted features (frequency and functional connectivity) have been used as input to deep learning and this increased the prediction accuracy by up to 7%. It also resulted in offering a tool for understanding how the EEG frequency and functional connectivity were affected by treatments across two groups (responder vs non-responder to tinnitus treatment).

We included the below information to page 9.

For example, Fig 5 shows that for a patient from non-responder group, the power of alpha frequency was increasing over time across most of the brain regions before the treatment and a similar pattern was seen after the treatment. On the other hand, Fig. 6 demonstrates that theta and alpha were shown to be more prominent after the treatment in the responder patient. These patterns of changes are used as inputs for training deep neural networks to distinguish who is likely to respond to treatment. 

We also included the below information to page 17.

The research proved that within EEG data, both frequency and functional connectivity contain significant information that shows brain changes as a response to treatment. Each of the frequency images and the functional connectivity graphs was generated using 1 second EEG signal (256 time points) and they were used as inputs to deep learning and achieved a greater accuracy of prediction compared to using the whole time points of raw EEG signals

********************************************

  • What is the real advancements in the field of the Journal. SInce there are no novelties in the filed of sensors one would expect some advancements in the field of signal processing.

This research does not focus on sensor development itself, but our findings suggest an optimal configuration of the EEG sensors that are involved in measuring the brain functional changes against tinnitus treatments.

The below information is included in page 3. 

Sensors have seldom been used clinically in relation to sensory disorders. It is novel to apply sensor technology to tinnitus and the algorithm and its application are innovative approaches to this complex sensory disorder.  The literature has clearly identified the need for biomarkers, and this is amongst the first studies looking towards effective wearable solutions in our field.

The below information is added to the abstract on page 1

The research suggests an optimal configuration of the EEG sensors that are involved in measuring the brain functional changes against tinnitus treatments. It identified which EEG electrodes are the most informative sensors and how the EEG frequency and functional connectivity can better classify patients into responder and non-responder groups.

.

3) another point that should be in depth discussed is the potential clinical impact of these results. 

On page 1, we stated that “Clinical judgements about the likelihood of treatment response are currently difficult, and patients are often switched through several treatments before finding an effective intervention. AI diagnostic prediction is an extremely promising new field in the treatment of tinnitus.”

On page 2, we also stated: AI may help in personalising the prescription approach to tinnitus, by indicating those interventions most likely to be effective for any given patient. This would improve clinical efficiency, reducing patient distress and optimising benefit.

Below information is included in page 3:

The clinical impact of the propped methodology is applying AI for early prediction of response to tinnitus treatment when only baseline data is used. This would allow an optimal selection of treatment options for patients through prediction of response to each treatment. We also identified a subset of EEG sensors as informative features that increase the accuracy prediction and make the further development of wearable diagnosis and prognosis AI tools more effective. Sensors have seldom been used clinically in relation to sensory disorders. It is novel to apply sensor technology to tinnitus and the algorithm and its application are innovative approaches to this complex sensory disorder.  The literature has clearly identified the need for biomarkers, and this is amongst the first studies looking towards effective wearable solutions in our field.

END ----------------------------------------

Reviewer 2 Report

The authors consider the task of helping people with tinnitus. Ringing in the ears is a hearing impairment characterized by the perception of sounds in the absence of an external source. There is currently no pharmaceutical cure for tinnitus, however, many treatments and interventions have been developed that improve or control the distress and anxiety associated with it. To evaluate these methods, it is proposed to use a new artificial intelligence algorithm as a digital health predictive system. The proposed algorithm differentiated patient outcomes with an accuracy of 98% to 100%.

1. The introduction should be extended. It is necessary to consider similar studies in more detail. Since one of the important tasks is the analysis and classification of EEG signals, more attention should be paid to this issue. It is possible to include some reviews on this topic: https://ieeexplore.ieee.org/document/9224666 and https://www.mdpi.com/1424-8220/21/24/8485

2. Could you explain the principal steps of methods for signal classification as feature extraction, feature selection and classification in your study and the algorithms which are used for their implementation?

3. Could you present in more detail the parameters of used classifiers (or feature selection methods?) in your paper: F_Regression (FR), Random Forest (RF), ExtraTrees (ET), and RFE? What do you mean as the procedure of feature selections? Is it possible to describe how these methods allow the selection of EEG signal features?

4. Is the result stable depending on the initial data? Do you expect the result to change if others samples of signals are used?

Author Response

Response to reviewers’ comments and summary of changes   

First of all, we would like to thank the reviewers for their very professional and time-consuming work to make this paper better organised and clearer, so that it will have a much stronger impact on the future research in our community.  It is novel to apply sensor technology to tinnitus and the algorithm and its application are innovative approaches to this complex sensory disorder.  We believe that our paper is a pioneering study in modelling EEG frequency domain and functional connectivity to predict response to tinnitus treatments. 

Below are the questions raised by the reviewers and our revision made in response.   

The authors consider the task of helping people with tinnitus. Ringing in the ears is a hearing impairment characterized by the perception of sounds in the absence of an external source. There is currently no pharmaceutical cure for tinnitus, however, many treatments and interventions have been developed that improve or control the distress and anxiety associated with it. To evaluate these methods, it is proposed to use a new artificial intelligence algorithm as a digital health predictive system. The proposed algorithm differentiated patient outcomes with an accuracy of 98% to 100%.

  1. The introduction should be extended. It is necessary to consider similar studies in more detail. Since one of the important tasks is the analysis andclassification of EEG signals, more attention should be paid to this issue. It is possible to include some reviews on this topic: https://ieeexplore.ieee.org/document/9224666 and https://www.mdpi.com/1424-8220/21/24/8485

Thank you for your comment and suggesting the literature. We have reviewed these papers and included our review in the revised version of our manuscript on page 2.

Modelling of high-dimensional EEG data using machine learning methods has been conducted for the diagnosis of various neurological diseases including epilepsy and seizures [41]. Research [42] proposed a Fuzzy Decision Tree (FDT) classifier for the epileptic's seizure detection. Its application allows achieving 99.5% accuracy of the classification of epileptic's seizure. Despite the existing of several successful EEG applications in health, there is still limited studies on modelling EEG for prediction of response to tinnitus treatment. Most of the EEG studies are based on classification of raw EEG signals using machine learning. Limited investigation was conducted on using dynamic functional connectivity and dynamic frequency images as inputs to deep learning for prediction of tinnitus treatment outcomes.

  1. Could you explain the principal steps of methods for signal classification as feature extraction, feature selection and classification in your study and the algorithms which are used for their implementation?

These main phases are explained in the below sections of the paper

2.3.1. EEG class prediction on frequency domain

2.3.2. EEG class prediction based on functional connectivity

  1. Feature Selection for identification of tinnitus predictive EEG variables

To improve the explanation, the following pages are modified:

Page 5 is modified as below:

As described in Equation (1) and [43], we first created EEG samples with 1s duration. As explained in Section. 2.1, 6,642 frequency images were generated. To make use of meaningful data, we selected features on the most prominent frequency bands (beta 14-30HZ, alpha 8-14HZ, theta 4-8HZ) as the analysis objects. The mean (across each 1 sec epoch) of absolute values for each of these frequencies was calculated at each electrode site.

 k=0,1,2….,N-1                     (1)

where  is the FFT coefficients, N is the total number of input EEG samples, n is total number of points in FFT.

Page 14 is modified as below:

The feature selection methods are called model-based ranking. We first calculated the average amplitude on each EEG channel (across patients) before and after treatment and obtained two (before and after treatment) 64×8 matrices, in which 64 represents the number of EEG channels, and 8 represents the number of patients. Then, another average was taken across 8 patients on each channel, thus generating two 64×1 matrices, one represents the EEG status (called before matric and after matric). We entered these two 64×1 matrices into the feature selection methods as described in the following.

For the FR method, the model returns the F-statistic and p-values as the criteria for ranking the importance of the channels. The larger the p-values, the higher the importance of the corresponding channel. This is calculated by measuring the correlation between before matric and after matric based on r-regression. Then, the cross correlation is converted to an F-score and then to a p-value.

The RF method ranks the importance of channels for each tree according to the impurity which is calculated based on the variance.

The ET method ranks the importance of the channels according to the impurity-based feature importance. The higher the rank, the more important the channel. The importance of a channel is computed as the (normalized) total reduction of the criterion brought by that channel. It is also known as the Gini importance.

The RFE method obtains the importance of each channel through the feature importance attribute returned by the model. Then, the least important channels are removed from the current set of channels. This step is repeated recursively on the channel set until the required number of channels is finally reached.

  1. Could you present in more detail the parameters of used classifiers (or feature selection methods?) in your paper: F_Regression (FR), Random Forest (RF), ExtraTrees (ET), and RFE? What do you mean as the procedure of feature selections? Is it possible to describe how these methods allow the selection of EEG signal features?

The parameters are now included in Table 6.

Also, more information on how these methods allow the selection of EEG features is included on page 14.

  1. Is the result stable depending on the initial data? Do you expect the result to change if others samples of signals are used?

The below statement is included in page 16.

In the process of model training, we used 70% of the data for training and the remaining 30% for network performance testing. We also conducted 5-fold cross-validation on the model, and the results were all above 90% accuracy. This suggest that our model is sufficiently robust and achieved high performances through multiple k-fold cross validation.

Round 2

Reviewer 2 Report

I thank authors for the consideration of comments. I have not other comments or recommendations